# The association between perceived sensitivity to medicines, reported side effects and personal characteristics: A secondary analysis of an RCT

**Alexandra Kern**[1]*, **Anja Fischbach-Zieger**[1], **Claudia Witt**[1,2,3], **Juergen Barth**[1]

**1** Institute for Complementary and Integrative Medicine, University Hospital Zurich and University of Zurich, Zurich, Switzerland, **2** Charité – Universitätsmedizin Berlin, Corporate Member of Freie Universität Berlin, Humboldt-Universität zu Berlin, Berlin, Germany, **3** Berlin Institute of Health, Institute of Social Medicine, Epidemiology and Health Economics, Berlin, Germany

* alexandra.kern@usz.ch

## Abstract

**Data Availability Statement:** All relevant data are within the manuscript.

### Introduction

A patient's perceived sensitivity to medicines (PSM) might influence the reported side effects of a treatment. The experience of side effects can result in personal and structural costs (such as nonadherence). Research on nocebo mechanisms and the workings of side effect reporting has been disproportionally smaller compared to the emerging evidence of the individual and clinical impact of the matter. In this study, we explored and re-examined the association between PSM and reported side effects within a specific population (chronic low back pain patients receiving acupuncture treatment), including possible mediating variables (i.e., gender, medical and nonmedical care utilization, optimism, pessimism, anxiety, depression and treatment expectation).

### Methods

We conducted a secondary analysis of a randomized controlled trial that examined the influence of treatment outcome expectations in chronic low back pain (CLBP) patients. All measures in the analysis were self-assessments. We explored the association between PSM, reported side effects and personal characteristics using correlational and regression analyses.

### Results

Our sample consisted of 152 CLBP patients (65.8% female), the mean age was 39.5 years (SD = 12.5). We found positive correlations between PSM and reported side effects (r = 0.24; p < 0.01) and between PSM and anxiety (r = 0.21; p < 0.05). However, the subsequent regression analyses did not confirm a mediating or moderating effect of anxiety between PSM and reported side effects.

**Funding:** The project was funded by the Swiss National Science Foundation (https://data.snf.ch/grants?q=159833), Grant Number 105319_159833. The funders had no role in study design, data collection and analysis, decision to publish, or preparation of the manuscript.

**Competing interests:** The authors have declared that no competing interests exist.

**Abbreviations: CLBP**, Chronic low back pain; **ETS**, Expectation for Treatment Scale; **HIV**, Human immunodeficiency viruses; **LOT-R**, Life Orientation Test–Revised; **PROMIS**, Patient-Reported Outcomes Measurement Information System®; **PSM**, Perceived Sensitivity to Medicines; **RCT**, Randomized controlled trial; **SD**, Standard deviation; **SPSS**, Statistical Package for the Social Sciences.

## Conclusion

We confirmed and extended earlier research on PSM. Our study involved a specific pain population receiving a nonpharmacological intervention. Our results highlight the importance of targeting PSM and anxiety within a treatment to take measures to mitigate the prevalence of side effects.

## Introduction

The nocebo phenomenon describes negative side effects that follow the administration of an inactive drug or treatment that the patient believes to be active [1]. Nocebo responses can also occur in the context of active treatments and elicit symptoms. This phenomenon exists in pharmacological treatments (i.e., chemotherapy) as well as in nonpharmacological treatments (i.e., acupuncture) [2, 3].

Regardless of whether the experience stems from a verum or placebo treatment, the experience of side effects can be uncomfortable or even frightening for an individual. Additionally, it can lead to undesired changes in the course of a treatment, such as nonadherence [4]. Therefore, the individual and structural costs of such a phenomenon are imminent. To date, research on contributing factors about side effect reporting (such as gender differences) and nocebo mechanisms has been disproportionally small compared to the emerging evidence of the individual and clinical impact of the matter [5, 6].

Known drivers of the nocebo response are classical conditioning and negative expectations toward a treatment [7]. It was empirically shown that treatment expectations play a vital role in the formation of nocebo and placebo responses [8]. This paradigm has been particularly well investigated in the area of pain processing including brain and behavioral markers [9]. However, certain aspects of expectation such as the development of expectations during the course of a treatment need some more consideration in studies to come [10].

Additionally, the personality of individuals might contribute to nocebo responses [11]. Furthermore, a negative affective state was considered relevant for nocebo responses in terms of symptom reporting [12]. Similarly, anticipatory anxiety was found to be associated with the nocebo response [13]. Geers and colleagues found that worry and depressive symptoms can trigger or worsen negative symptoms, especially with respect to vaccine side effects [14]. Moreover, there has been a series of studies suggesting that gender might account for differences in nocebo responses [15], which is in line with studies showing that side effects from medication are in general more frequent among women [16]. On this basis, Barsky and colleagues demand that further emphasis be placed on personal characteristics that increase the likelihood of experiencing side effects [1].

The belief that a person is distinctively sensitive to the (side) effects of a treatment can be regarded as one of the potential drivers of the nocebo response. The so-called perceived sensitivity to medicines (PSM) can act as a precursor to forming negative expectations about a treatment and therefore represents a substantial driver of nocebo responses [6]. For the assessment of PSM, Horne and colleagues developed a validated PSM scale with good reliability [17]. The scale consists of five self-rating items, including statements such as "my body is very sensitive to medicines" or "my body overreacts to medicines". High PSM is regarded as relatively common in the general population. In line with Kardas' and Petrie's suggestions, higher PSM was associated with increased symptom reporting, a higher number of medical visits [18] and an increased risk of nonadherence [19]. In a more recent study, Svensberg and colleagues

supported the notion that high PSM is common among the general population. Furthermore, female gender was positively associated with high PSM, whereas lower educational level was also associated with higher PSM [20].

In our study, we used acupuncture as an intervention. Side effects of acupuncture are prevalent albeit often minor (such as hematoma or headache). Serious adverse events associated with acupuncture, such as pneumothorax, are very rare [21]. Acupuncture can serve as an ideal paradigm to investigate nocebo responses since minor side effects are prevalent with this treatment.

Our study aimed to explore the association between PSM and reported side effects within a population of chronic low back pain (CLBP) patients receiving acupuncture treatment. Our goal was to evaluate the association between PSM and reported side effects and to find possible mediating variables.

## Methods

We conducted a secondary analysis of a randomized controlled trial (RCT) that evaluated the influence of treatment outcome expectations in CLBP patients. This detailed procedure has been described earlier [22] and in the trial registration (German Clinical Trial Register): DRKS00010191. The trial was approved by the cantonal ethics committee of Zurich. All participants gave written informed consent.

### Participants

The sample consisted of patients suffering from CLBP. The sample size was 152, which refers to the number of patients needed in the RCT. Due to listwise deletion of incomplete (missing) data in the dependent variables, the number of patients included in the regression model was 148 patients. The recruiting process used in this study is described in detail in an earlier publication [22].

### Intervention

All CLBP patients received the same standardized minimal acupuncture treatment, which consisted of eight acupuncture sessions. Patients were randomized according to a 2 x 2 factorial design. They received either a regular expectation briefing or a high expectation briefing and either a regular adverse side effect briefing or an intense adverse side effect briefing. The standardized briefings were given in person by a trained physician and in written form. We found no effect of the type of briefing on reported side effects [22].

### Measures

All measures for this analysis are self-assessments from baseline (i.e., with a baseline assessment made prior to the first acupuncture session and the expectation briefing) except for the reported side effects. Side effects, as a patient-reported outcome, were assessed repeatedly before each acupuncture treatment from session two to session eight.

### Perceived sensitivity to medicines

The Perceived Sensitivity to Medicines (PSM) Scale is considered a reliable and valid measure composed of five self-report questions to assess perceived sensitivity to the potential adverse effects of medicines [17]. Responses are scored on a five-point Likert scale, and patients' item scores are summed to provide a total score ranging between 5 and 25. Higher scores indicate a

high perceived sensitivity to the potential adverse effects of medicines. In the current study, the scale showed excellent internal consistency (Cronbach's alpha = 0.91).

### Reported side effects

The patients evaluated their reported side effects in retrospect before every acupuncture session. Thirteen side effects were listed and could be rated each from 0 (not at all) to 3 (severe). For each person, a session score was calculated by multiplying the occurrence of the side effect (e.g., headache or the occurrence of a hematoma) and the severity. The total sum score was calculated by summing the scores from seven sessions (range from 0 to 273) and used for the analysis.

### Medical care utilization and nonmedical care utilization

Past medical and nonmedical care utilization (i.e., prior to the initial acupuncture session) was part of the baseline questionnaire. Patients were asked to indicate if they had used medical care (e.g., rheumatologist) or nonmedical care (e.g., massage) due to their low back pain during the previous eight weeks. The format of the questions was binary (yes/no).

### Optimism and pessimism

Both concepts were assessed using the German version of the Life Orientation Test–Revised (LOT-R) [23]. The questionnaire consists of six self-reported items (plus four filler items), each rated on a five-point Likert scale ranging from 0 (strongly disagree) to 4 (strongly agree). Data were separated into optimism and pessimism scores, as recommended [23]. Each score can range from 0 to 12, with higher values indicating either higher optimism or pessimism. In the present study, Cronbach's alpha was acceptable for optimism (0.74) and questionable for pessimism (0.69).

### Depression and anxiety

Self-reported health was assessed with the 29-item short-form of the Patient-Reported Outcomes Measurement Information System® (PROMIS; www.nihpromis.org). The PROMIS Profile-29 combines the 4-item short forms from seven PROMIS domains (depression, anxiety, physical function, pain interference, fatigue, sleep disturbance, and ability to participate in social roles and activities) and a single item on pain intensity [24]. The items of each domain are rated on a five-point Likert scale, and the total score of each domain was converted to a standardized T-score. Higher scores for negatively phrased concepts (anxiety, depression, pain interference, fatigue, sleep disturbance) and lower scores for positively phrased concepts (physical function and ability to participate in social roles and activities) indicate less self-reported health. In the current study, we used the anxiety and depression scale for our analysis. Cronbach's alpha was good for both the depression scale (0.88) and for the anxiety scale (0.84).

### Expectation for treatment scale

Treatment expectation was assessed using the Expectation for Treatment Scale (ETS), which consists of five items that capture the personal evaluation of each patient regarding whether a specific treatment might be beneficial for their complaints [25]. Each item was to be rated on a four-point scale ranging from 1 to 4 (partially disagree, partially agree, agree, definitely agree), with a total score from 5 to 20. In the present study, Cronbach's alpha was acceptable (0.77).

## Analysis

Since our analysis includes a multitude of possible mediating variables, we registered our study at Aspredicted.org. (Registration number 27734, created 09/11/2019). Data were analyzed using SPSS (SPSS version 25.0; IBM Corp., Armonk, NY, USA). To examine the normal distribution of our data, we used the Shapiro Wilk test. In cases where the assumption of a normal distribution was not met, we log transformed our data. To determine the association between our main variables (PSM and side effects) and our potentially mediating variables (gender, medical and nonmedical care utilization, optimism, pessimism, anxiety, depression and treatment expectation), we used correlational analyses. We calculated Pearson correlation coefficients for our parametric data and Spearman correlation coefficients for our nonparametric data (i.e., gender, medical and nonmedical care utilization). In a further step, we calculated mediator and moderator analyses for the variables that showed a significant correlation with PSM using a linear regression model. To enhance the interpretability of our main effects, we mean-centered values in the mediation and moderation analyses. For the mediation and moderation analyses, we followed the traditional steps outlined by Baron and Kenny [26]. In a first step, the correlation between the independent variable (i.e., PSM) and the depended variable (i.e., reported side effects) was tested prior to further meditation or moderation analyses. In a second step, mediation analyses were conducted in order to determine the extent to which the possible mediator accounts for the relationship between the independent and the dependent variable. In a last step, moderation analyses were conducted in order to detect the direction/strength of the relationship between the possible moderator, the independent and the dependent variable.

## Results

### Patients' demographics and clinical characteristics

A total of 152 patients were included in the analysis (65.8% female), and the mean age was 39.5 years (SD = 12.5). The mean pain intensity was rated 5.4 (SD = 1.3), and the mean score of the PSM was 11.0 (SD = 5.1). Two-thirds (67.8%) of the patients indicated that they used medical care for CLBP in the past, whereas only 44.7% used nonmedical care for their pain. More information on other characteristics can be found in Table 1.

### Correlations and regression models

Shapiro–Wilk tests showed that none of the metric variables were normally distributed, which led to a log transformation of all variables for subsequent analyses.

We found a positive correlation between PSM and reported side effects (r = 0.24; p < 0.01). Table 2 shows that PSM correlated positively with anxiety (r = 0.21; p < 0.05) and that gender correlated negatively with reported side effects (r = − 0.25; p < 0.01). No other statistically meaningful correlations were found in the total sample. In the female subsample, we found a negative correlation between PSM and medical utilization (r = − 0.20; p < 0.05), as shown in Table 3. Tables with pairwise correlations can be found in the supplementary tables.

In an additional sensitivity analysis we looked at the side effect score of the first four sessions. Our analysis revealed the same baseline variables as predictors of side effects compared to the predictive baseline variables across all eight sessions.

Although we found a significant correlation between PSM and anxiety, there was no significant correlation between anxiety and the reported side effects. Our regression model included reported side effects as the dependent variable and PSM and anxiety as the independent variables. The overall model was significant (F(2) = 4.44, p = 0.014, $R^2$ = 0.06). PSM was associated

**Table 1. Sample baseline characteristics.** Values are numbers (percentages) or means (standard deviation), N = 152.

| | |
|---|---|
| Gender | |
| Female | 100 (65.8%) |
| Male | 52 (34.2%) |
| Age, yrs | 39.54 (12.54) |
| Education | |
| School ≤ 10 years | 3 (2%) |
| School ≥ 12 years | 86 (56.6%) |
| University | 63 (41.1%) |
| Pain intensity (NRS) | 5.37 (1.33) |
| Perceived sensitivity to medicines (PSM) | 11.01 (5.14) |
| Medical care utilization (past 8 weeks) | |
| Yes | 103 (68.2%) |
| No | 48 (31.8%) |
| Nonmedical care utilization (past 8 weeks) | |
| Yes | 68 (45.3%) |
| No | 82 (54.7%) |
| Expectation (ETS) | 12.12 (3.24) |
| Self-reported health (PROMIS) | |
| Anxiety | 54.57 (8.01) |
| Depression | 52.29 (8.01) |
| Ability to participate in social roles | 49.01 (7.55) |
| Fatigue | 54.15 (9.12) |
| Pain interference | 58.38 (4.74) |
| Physical functioning | 45.15 (5.53) |
| Sleep disturbance | 51.45 (8.74) |
| Personality | |
| Optimism (LOT-R) | 8.74 (2.48) |
| Pessimism (LOT-R) | 3.42 (2.30) |

ETS = Expectation for Treatment Scale; LOT-R = Life Orientation Test Revised; NRS = Numeric Rating Scale; PSM = Perceived Sensitivity to Medicines; PROMIS = Patient-Reported Outcomes Measurement Information System

**Table 2. Correlations between independent variable (PSM), dependent variable (reported side effect score) and variables in the mediation model N = 152.**

| | PSM | Reported side effect score |
|---|---|---|
| Gender | -0.13 | -0.25** |
| Medical care utilization | -0.08 | -0.02 |
| Nonmedical care utilization | 0.06 | -0.04 |
| LOT-R optimism | -0.11 | -0.07 |
| LOT-R pessimism | 0.14 | -0.02 |
| PROMIS depression | 0.12 | 0.11 |
| PROMIS anxiety | 0.21* | 0.09 |
| ETS treatment expectation | 0.04 | -0.03 |

**Correlation is significant at the 0.01 level (2-tailed)

*Correlation is significant at the 0.05 level (2-tailed)

ETS = Expectation for Treatment Scale, LOT-R = Life Orientation Test Revised, PSM = Perceived Sensitivity to Medicines, PROMIS = Patient-Reported Outcome Measurement Information System

Coding: gender = male/female, medical care utilization = yes/no, non-medical care utilization = yes/no

**Table 3. Correlations between independent variable (PSM), dependent variable (reported side effect score) and variables in the mediation model for females only, N = 100.**

|  | PSM | Reported side effect score |
|---|---|---|
| Gender | - | - |
| Medical care utilization | -0.20* | 0.03 |
| Nonmedical care utilization | 0.10 | -0.10 |
| LOT-R optimism | -0.05 | -0.10 |
| LOT-R pessimism | 0.10 | -0.05 |
| PROMIS depression | 0.05 | 0.06 |
| PROMIS anxiety | 0.15 | 0.04 |
| ETS treatment expectation | 0.06 | -0.06 |

\*\*Correlation is significant at the 0.01 level (2-tailed)

\*Correlation is significant at the 0.05 level (2-tailed)

ETS = Expectation for Treatment Scale, LOT-R = Life Orientation Test Revised, PSM = Perceived Sensitivity to Medicines, PROMIS = Patient-Reported Outcome Measurement Information System

Coding: gender = male/female, medical care utilization = yes/no, non-medical care utilization = yes/no

with reported side effects (β = 0.24; p = 0.003), while anxiety was independent of reported side effects (β = 0.09; p = 0.271). According to Baron and Kenny [26], the requirements for a mediating effect were not met due to the independence between anxiety and reported side effects. The regression model for the overall moderation analysis was significant (F(3) = 3.84, p = 0.011, $R^2$ = 0.07), indicating that the included variables were able to explain the reported side effects. However, none of the beta weights of PSM, anxiety and the interaction term of PSM and anxiety were significant alone.

## Discussion

Overall, we found positive correlations between PSM and reported side effects and between PSM and anxiety within a population of CLBP patients receiving acupuncture treatment. However, the subsequent regression analyses did not confirm a mediating or moderating effect of anxiety between PSM and reported side effects.

Our results support the former finding by Faasse and colleagues that high PSM is associated with increased symptom reporting. In contrast to this former study, we could not confirm the finding that high PSM is associated with increased medical visits [18]. Whereas Faasse et al. found that patients with high PSM reported more general practitioner visits, our results showed the opposite association in the female subsample. It is important to note that our results are focused on one specific disorder only (namely, CLBP), whereas the aforementioned study was based on a healthy general population cohort. Furthermore, we used a broader definition of medical care utilization than Faasse and colleagues did.

Concerning the associated factors of PSM and patient characteristics, some of our findings are in line with previous studies, while others are not. Our results showed an association between PSM and anxiety. However, our results did not show an association between anxiety and side effect reporting. The latter finding is in contrast with the results of the study by Petrie et al., which found an association between negative affect and increased symptom reporting [12]. Unexpectedly, we did not find an association between gender and PSM. Nor did we find an association between gender and increased side effect reporting. Both results are in contrast with former findings. Zopf and colleagues found that women showed more side effects from medication than men [16]. A secondary data analysis found that female gender was associated

with higher odds of reporting an adverse reaction in randomized acupuncture trials [27]. Svensberg et al. found female gender to be associated with higher PSM [20]. Especially because our results are not in line with former findings, the role of gender or sex in placebo and nocebo effects are still not clear cut [28], and the assessment of contributing factors is greatly warranted to gain comprehensible insights in the future [29].

The PSM questionnaire is easy to administer for patients and care providers. Kalichman et al. found PSM to be directly related to the experience of side effects, which in turn predicted intentional and overall nonadherence in HIV patients [30]. Consequently, the authors argue that patients with high PSM are at greater risk for poor treatment outcomes. The prescreening of patients on their beliefs about medicines to identify those who are at high risk of nonadherence could be recommended to tackle this issue [19]. Our study provides an addition to these results since we were able to confirm the association between PSM and side effects even beyond pharmacological interventions. In sum, the importance of carefully assessing preexisting beliefs such as PSM is warranted. Besides personal characteristics, it seems worthwhile to take contextual factors (such as the relationship between the caregiver and the patient or the health care setting) into account. Paying attention to these factors might help offset the nocebo response [31, 32].

Among the strengths of our study are the large sample size and the broad number of potentially influencing factors in the model. The latter also represents an advantage to our exploratory aim of this study based on a preregistered selection of contributing variables. Moreover, the applied clinical setting of the study contributes to a better translation of the results into clinical practice. Rossettini and colleagues express in their state of the art paper that more studies on the effects of placebo and nocebo in chronic pain patients are needed as opposed to acute pain patients [33]. We regard our study as a contribution in closing this gap.

The larger proportion of female patients in the sample can be seen as a limitation of our study. Additionally, the sample was highly educated, which may affect the findings since a lower educational level may be associated with higher PSM [20].

## Conclusion

With our study, we were able to confirm and extend earlier research on PSM by incorporating a nonpharmacological intervention in a specific pain population. We found positive correlations between PSM and reported side effects and between PSM and anxiety. Our results indicate that targeting anxiety and the perception of individual sensitivities to medicines before a treatment may help to mitigate the prevalence of side effects.

## Supporting information

**S1 Table.**
(DOCX)

## Author Contributions

**Formal analysis:** Alexandra Kern.

**Project administration:** Alexandra Kern, Anja Fischbach-Zieger.

**Supervision:** Claudia Witt, Juergen Barth.

**Writing – original draft:** Alexandra Kern.

**Writing – review & editing:** Anja Fischbach-Zieger, Claudia Witt, Juergen Barth.

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
