## [Decision Letter · Decision Letter 0]

19 Dec 2023

PONE-D-23-31531The association between perceived sensitivity to medicines, reported side effects and personal characteristicsPLOS ONE

Dear Dr. Kern,

Thank you for submitting your manuscript to PLOS ONE. After careful consideration, we feel that it has merit but does not fully meet PLOS ONE’s publication criteria as it currently stands. Therefore, we invite you to submit a revised version of the manuscript that addresses the points raised during the review process.

We note that one or more reviewers has recommended that you cite specific previously published works. As always, we recommend that you please review and evaluate the requested works to determine whether they are relevant and should be cited. It is not a requirement to cite these works. 

We look forward to receiving your revised manuscript.

Kind regards,

Vanessa Carels

Staff Editor

PLOS ONE

2. We note that your Data Availability Statement is currently as follows: [All relevant data are within the manuscript.]

Reviewers' comments:

Reviewer's Responses to Questions

**Comments to the Author**

1. Is the manuscript technically sound, and do the data support the conclusions?

Reviewer #1: Yes

Reviewer #2: Partly

2. Has the statistical analysis been performed appropriately and rigorously? 

Reviewer #1: Yes

Reviewer #2: No

3. Have the authors made all data underlying the findings in their manuscript fully available?

Reviewer #1: Yes

Reviewer #2: Yes

4. Is the manuscript presented in an intelligible fashion and written in standard English?

Reviewer #1: Yes

Reviewer #2: Yes

5. Review Comments to the Author

Reviewer #1: Dear Authors,

Thanks a lot for the opportunity you have offered me to revise the fascinating manuscript " The association between perceived sensitivity to medicines, reported side effects and personal characteristics". I thank the authors for their efforts in producing this exciting manuscript. It perfectly aligns with my research and expertise; thus, I am confident I can offer a valuable peer review.

As a significant strength, this manuscript aims to explore and re-examined the association between PSM and reported side effects within a specific population (chronic low back pain patients receiving acupuncture treatment), including possible mediating variables (i.e., gender, medical and nonmedical care utilization, optimism, pessimism, anxiety, depression and treatment expectation). This proposal is really interesting in the field and adds information to the existing evidence in the literature.

As a major weakness, the manuscript sometimes lacks details and clarity concerning methodological and content steps that would help improve the understanding of the manuscript. Therefore, I have suggested some strategies to improve authors' reporting and increase the quality of their work.

Overall, my peer review is a minor revision: I suggest revising the manuscript to improve the pitfalls presented. The final goal is to improve the overall clarity of the message to help the reader understand this fundamental topic. I look forward to reading the revised version of the manuscript.

Thanks again, and good luck with researching in this challenging time.

¶MINOR REVISION

#TITLE:

*add the study design to the title: “characteristics: a secondary analysis of a RCT”

#ABSTRACT:

*Results: please add the mean age of participants in the brackets ().

#RESULTS:

*structure: I suggest authors split the results section into different sections. For example, divide the findings related to the patient’s demographic from the data related to the correlations. It would help readers to follow the flow better.

¶MAJOR REVISION

#INTRODUCTION

*Background: The authors analyse the concepts of nocebo and patient experience of side effects. Although their work is commendable, it is appropriate to integrate some fundamental references for nocebo into the background, focusing on patients with musculoskeletal pain (doi: 10.3390/jcm12124113 doi: 10.1016/j.msksp.2022.102677). I suggest reading and adding these references in the background.

#DISCUSSION:

*expectations and nocebo: The authors have really well developed the discussion. They should develop the role of negative expectations in the perception of adverse effects in patients with musculoskeletal pain. This topic is an emerging hot topic that should be considered, especially for the clinical implications they have and can be read from the perspective of brain predictive mechanisms. Indeed, when expectations are negative, they are sometimes difficult to change. In this regard, I suggest reading and integrating this recent review on the topic (doi: 10.3389/fpsyg.2022.789377).

*implications for clinical practice: the study’s findings are indeed relevant, but it is necessary to emphasise the clinical implications. Indeed, clinicians need to become aware of nocebo in their clinical practice to improve the effect of their therapies, reduce the risk of drop-outs and enhance the therapeutic outcome. This clearly emerges from several cross-sectional studies conducted on nurses (doi: 10.1111/jocn.14809), physiotherapists (doi: 10.3389/fpsyg.2020.582174) and other healthcare professionals (doi: 10.1371/journal.pone.0291079) who manage patients with musculoskeletal pain daily around the world. Moreover, this finding is all the more important given that patients with musculoskeletal pain want their expectations to be properly considered (doi: 10.3389/fpsyt.2019.00478). A reading of these studies and a reflection on the topic is worthwhile.

Reviewer #2: This report is a secondary analysis of a randomized controlled trial (DRKS00010191) dataset focusing on the association between perceived sensitivity to medicines (PSM) and reported side effects, and whether the association was mediated or moderated by other variables, in particular, anxiety. The authors already published another secondary data analysis paper from the same trail in the journal (https://doi.org/10.1371/journal.pone.0268646).

1. The logic, method, and criteria of mediation and moderation analysis should be clearly described in the Methods section, and then report the results in the Result section. Currently the methods were not described in the correct place.

2. On Tables 2 and 3, I suggest making correlation matrice / heatmaps to show pairwise correlations of all variables to make the results more informative.

3. As the side effect score is the dependent variable, and gender is significantly associated with it, it’s strange the mediation and moderation analyses for gender were not performed.

4. The current analysis is limited to perform mediation / moderation analysis for variables with significant correlation with the (in)dependent variables. I suggest trying a variable selection to find out what you get.

5. The side effects score was a summary of all sessions. I suggest performing a sensitivity analysis by looking at the side effects scores at earlier and later sessions, similar to the strategy in the other PLoS ONE paper you published. A hypothesis is the distribution of side effects scores, or its correlation with other variables, were changed.

6. PLOS authors have the option to publish the peer review history of their article (what does this mean?). If published, this will include your full peer review and any attached files.

Reviewer #1: No

Reviewer #2: No

---

## [Author Response · Author response to Decision Letter 0]

29 Mar 2024

Reviewer #1: Dear Authors,

Thanks a lot for the opportunity you have offered me to revise the fascinating manuscript; The association between perceived sensitivity to medicines, reported side effects and personal characteristics;. I thank the authors for their efforts in producing this exciting manuscript. It perfectly aligns with my research and expertise; thus, I am confident I can offer a valuable peer review.

As a significant strength, this manuscript aims to explore and re-examined the association between PSM and reported side effects within a specific population (chronic low back pain patients receiving acupuncture treatment), including possible mediating variables (i.e., gender, medical and nonmedical care utilization, optimism, pessimism, anxiety, depression and treatment expectation). This proposal is really interesting in the field and adds information to the existing evidence in the literature.

As a major weakness, the manuscript sometimes lacks details and clarity concerning methodological and content steps that would help improve the understanding of the manuscript. Therefore, I have suggested some strategies to improve authors’ reporting and increase the quality of their work.

Overall, my peer review is a minor revision: I suggest revising the manuscript to improve the pitfalls presented. The final goal is to improve the overall clarity of the message to help the reader understand this fundamental topic. I look forward to reading the revised version of the manuscript.

Thanks again, and good luck with researching in this challenging time.

Response: Dear Reviewer #1, we kindly thank you for your helpful and in-depth review of our manuscript. Please find our responses and corresponding changes below.

MINOR REVISION

#TITLE:

*add the study design to the title: characteristics: a secondary analysis of a RCT

Response: We changed the title accordingly. It now reads: The association between perceived sensitivity to medicines, reported side effects and personal characteristics: a secondary analysis of an RCT.

#ABSTRACT:

*Results: please add the mean age of participants in the brackets ().

Response: We added the mean age of participants. It now says: Our sample consisted of 152 CLBP patients (65.8% female), the mean age was 39.5 years (SD = 12.5).

#RESULTS:

*structure: I suggest authors split the results section into different sections. For example, divide the findings related to the patient’s demographic from the data related to the correlations. It would help readers to follow the flow better.

Response: We split the result section into patients’ demographics and clinical characteristics and correlations and regression models in the revised manuscript

MAJOR REVISION

#INTRODUCTION

*Background: The authors analyse the concepts of nocebo and patient experience of side effects. Although their work is commendable, it is appropriate to integrate some fundamental references for nocebo into the background, focusing on patients with musculoskeletal pain (doi: 10.3390/jcm12124113 doi: 10.1016/j.msksp.2022.102677). I suggest reading and adding these references in the background.

Response: Thank you for the valuable reference. Since the mentioned first paper concerns the aspect of chronic pain (as opposed to acute pain), we included the reference in the discussion section where it fits very well. It now says: Rossettini and colleagues express in their state of the art paper that more studies on the effects of placebo and nocebo in chronic pain patients are needed as opposed to acute pain patients [33]. We regard our study as a contribution in closing this gap.

The topic of the second paper on the patient-caregiver relationship is incorporated further below in the implications section.

#DISCUSSION:

*expectations and nocebo: The authors have really well developed the discussion. They should develop the role of negative expectations in the perception of adverse effects in patients with musculoskeletal pain. This topic is an emerging hot topic that should be considered, especially for the clinical implications they have and can be read from the perspective of brain predictive mechanisms. Indeed, when expectations are negative, they are sometimes difficult to change. In this regard, I suggest reading and integrating this recent review on the topic (doi: 10.3389/fpsyg.2022.789377).

Response: Thank you for your thoughts on expectations. Since the topic is closely related to the next comment, we decided to merge the two comments and subsequently integrate them into the discussion section. Please see below.

*implications for clinical practice: the study’s findings are indeed relevant, but it is necessary to emphasise the clinical implications. Indeed, clinicians need to become aware of nocebo in their clinical practice to improve the effect of their therapies, reduce the risk of drop-outs and enhance the therapeutic outcome. This clearly emerges from several cross-sectional studies conducted on nurses (doi: 10.1111/jocn.14809), physiotherapists (doi: 10.3389/fpsyg.2020.582174) and other healthcare professionals (doi: 10.1371/journal.pone.0291079) who manage patients with musculoskeletal pain daily around the world. Moreover, this finding is all the more important given that patients with musculoskeletal pain want their expectations to be properly considered (doi: 10.3389/fpsyt.2019.00478). A reading of these studies and a reflection on the topic is worthwhile.

Response: We thank the reviewer for suggestions of additional studies. While all of the mentioned references are relevant for a variety of clinical fields and very interesting to read, we selected the studies, which are most impactful for our specific topic. We decided to elaborate on contextual factors in general and leave out studies that do not directly touch our topic, such as the study concerning orthopedic manual therapists (OMTs).

We added the following to the implications sections: Besides personal characteristics, it seems worthwhile to take contextual factors (such as the relationship between the caregiver and the patient or the health care setting) into account. Paying attention to these factors might help offset the nocebo response [31, 32].

Reviewer #2: This report is a secondary analysis of a randomized controlled trial (DRKS00010191) dataset focusing on the association between perceived sensitivity to medicines (PSM) and reported side effects, and whether the association was mediated or moderated by other variables, in particular, anxiety. The authors already published another secondary data analysis paper from the same trail in the journal (https://doi.org/10.1371/journal.pone.0268646).

1. The logic, method, and criteria of mediation and moderation analysis should be clearly described in the Methods section, and then report the results in the Result section. Currently the methods were not described in the correct place.

Response: Thank you for this comment. We added the following methodological considerations about mediation and moderation in the methods section: For the mediation and moderation analyses, we followed the traditional steps outlined by Baron and Kenny [26]. In a first step, the correlation between the independent variable (i.e., PSM) and the depended variable (i.e., reported side effects) was tested prior to further meditation or moderation analyses. In a second step, mediation analyses were conducted in order to determine the extent to which the possible mediator accounts for the relationship between the independent and the dependent variable. In a last step, moderation analyses were conducted in order to detect the direction/strength of the relationship between the possible moderator, the independent and the dependent variable.

All results are still presented in the results section; however, the explanation of the methodology was shifted as suggested.

2. On Tables 2 and 3, I suggest making correlation matrice / heatmaps to show pairwise correlations of all variables to make the results more informative.

Response: Thank you for this input. We added two pairwise correlations tables of all variables to the appendix. See appendix a and b.

3. As the side effect score is the dependent variable, and gender is significantly associated with it’s strange the mediation and moderation analyses for gender were not performed.

Response: Thank you for your valuable comment.

In the case of the mediation analysis, we refer to our response number 4. Since there was no correlation between gender and the independent variable (PSM) in the first place, we refrained from conducting a mediation analysis for gender, PSM and reported side effect scores.

However, as you have correctly noted, there is a negative correlation between gender and the dependent variable (r = - 0.25; p < 0.01). Which allows us – according to the definition of Barron and Kenny (1986) - to run a moderation analysis. Even though the interpretation of such a statistical approach is rather limited.

We subsequently conducted a moderation analysis for PSM, reported side effect scores and the moderator (representing the interaction term between PSM and gender). 

Results showed an adjusted R-squared of 0.10. The overall model was significant, F(3) = 6.50, p < 0.001. However, the interaction term between PSM and gender was not significant in the ANOVA model.

4. The current analysis is limited to perform mediation / moderation analysis for variables with significant correlation with the (in)dependent variables. I suggest trying a variable selection to find out what you get.

Response: Concerning the mediation and moderation analyses we decided to orient ourselves by means of the traditional outline of mediation and moderation analyses by Baron and Kenny (1986). We assume three paths: a, b and c. Path c being the direct impact of the independent variable (i.e., PSM) on the dependent variable (i.e., reported side effects). Path b being the impact of the mediator on the dependent variable. Path a being the path from the independent variable to the mediator.

According to Barron and Kenny a variable classifies as a mediator if the following conditions are met:

1. Variations in levels of the independent variable (in our case PSM) significantly accounts for variations in the presumed mediator (path a). 2. Variations in the mediator significantly account for variations in the dependent variable (in our case reported side effects) (path b). 3. When path and b are controlled, a previously significant relation between the independent and dependent variables is no longer significant (path c).

We conducted a mediation analysis for anxiety as a presumed mediator since anxiety was the only variable that met criterion 1 as listed above. 

Baron, R. M., & Kenny, D. A. (1986). The moderator–mediator variable distinction in social psychological research: Conceptual, strategic, and statistical considerations. Journal of Personality and Social Psychology, 51(6), 1173-1182. DOI:10.1037/0022-3514.51.6.1173

5. The side effects score was a summary of all sessions. I suggest performing a sensitivity analysis by looking at the side effects scores at earlier and later sessions, similar to the strategy in the other PLoS ONE paper you published. A hypothesis is the distribution of side effects scores, or its correlation with other variables, were changed.

Response: Thank you for this complementation. We conducted additional analyses by using the side effect scores after session 4 and compared the findings with the already presented results after session 8. The results for session 4 were the following:

Distribution of the side effect scores: Both tests for normal distribution (Kolmogorov-Smirnov and Shapiro-Wilk test) did not show normal distribution of the data. Again, a log transformation of the side effect scores was done prior to further correlational analyses.

Correlation between side effect scores after session 4 and other variables: No substantial additional correlations (compared to session 8) between side effect scores after session 4 and other variables emerged. The correlation between PSM and side effect scores after session 4 was r = 0.20, p < 0.05. The correlation between side effect scores after session 4 and gender was not significant, r = - 0.13, p > 0.05. The latter result supports our quest of not continuing to pursue further moderation analyses between PSM, gender and side effect scores.

---

## [Decision Letter · Decision Letter 1]

16 May 2024

PONE-D-23-31531R1The association between perceived sensitivity to medicines, reported side effects and personal characteristics: a secondary analysis of an RCTPLOS ONE

Dear Dr. Kern,

Thank you for submitting your manuscript to PLOS ONE. After careful consideration, we feel that it has merit but does not fully meet PLOS ONE’s publication criteria as it currently stands. Therefore, we invite you to submit a revised version of the manuscript that addresses the points raised during the review process.

Your revision should address carefully the comments of referee #2, with which I totally concur. Although your paper has improved, it requires that you incorporate changes implied by your response letter as described in the comments below.

We look forward to receiving your revised manuscript.

Kind regards,

Mireia Jofre-Bonet, PhD

Academic Editor

PLOS ONE

Journal Requirements:

Reviewers' comments:

Reviewer's Responses to Questions

**Comments to the Author**

1. If the authors have adequately addressed your comments raised in a previous round of review and you feel that this manuscript is now acceptable for publication, you may indicate that here to bypass the “Comments to the Author” section, enter your conflict of interest statement in the “Confidential to Editor” section, and submit your "Accept" recommendation.

Reviewer #1: All comments have been addressed

Reviewer #2: (No Response)

2. Is the manuscript technically sound, and do the data support the conclusions?

Reviewer #1: Yes

Reviewer #2: Yes

3. Has the statistical analysis been performed appropriately and rigorously? 

Reviewer #1: Yes

Reviewer #2: Yes

4. Have the authors made all data underlying the findings in their manuscript fully available?

Reviewer #1: Yes

Reviewer #2: No

5. Is the manuscript presented in an intelligible fashion and written in standard English?

Reviewer #1: Yes

Reviewer #2: Yes

6. Review Comments to the Author

Reviewer #1: Congratulation. The paper is ready for the publication.

I am more than happy to accept it.

Please continue to write this interesting ma uscito.

Reviewer #2: The authors are very responsive on the critiques. However, some responses are only in the rebuttal letter, but not reflected in the manuscript. Please keep in mind the audience don’t get the chance to read to rebuttal letter. This is particular relevant to my previous comment 5, a sensitivity analysis by looking at the side effects scores at earlier and later sessions.

On the pairwise correlation tables, first it is more proper to be named Supplementary tables than appendices since there is analysis involved; second, it is not referred to in the main text.

7. PLOS authors have the option to publish the peer review history of their article (what does this mean?). If published, this will include your full peer review and any attached files.

Reviewer #1: No

Reviewer #2: No

---

## [Author Response · Author response to Decision Letter 1]

26 Jun 2024

Response to Reviewers

Reviewer #1: Congratulation. The paper is ready for the publication.

I am more than happy to accept it.

Please continue to write this interesting ma uscito.

Response: Thank you very much for diligently reviewing the manuscript and for your words of encouragement.

Reviewer #2: The authors are very responsive on the critiques. However, some responses are only in the rebuttal letter, but not reflected in the manuscript. Please keep in mind the audience don’t get the chance to read to rebuttal letter. This is particular relevant to my previous comment 5, a sensitivity analysis by looking at the side effects scores at earlier and later sessions.

Response: Thank you for this valuable note. In consequence, we included a comment on the sensitivity analysis in the result section of the manuscript, which reads: In an additional sensitivity analysis we looked at the side effect score of the first four sessions. Our analysis revealed the same baseline variables as predictors of side effects compared to the predictive baseline variables across all eight sessions.

On the pairwise correlation tables, first it is more proper to be named Supplementary tables than appendices since there is analysis involved; second, it is not referred to in the main text.

Response: This is correct, thank you. It is indeed a supplement not an appendix. We renamed the file accordingly. Furthermore, we referred to the tables in the result section of the manuscript as follows: Tables with pairwise correlations can be found in the supplementary tables.

---

## [Editor Report · Decision Letter 2]

19 Jul 2024

The association between perceived sensitivity to medicines, reported side effects and personal characteristics: a secondary analysis of an RCT

PONE-D-23-31531R2

Dear Dr. Kern,

We’re pleased to inform you that your manuscript has been judged scientifically suitable for publication and will be formally accepted for publication once it meets all outstanding technical requirements.

Kind regards,

Mireia Jofre-Bonet, PhD

Academic Editor

PLOS ONE

---

## [Editor Report · Acceptance letter]

22 Aug 2024

PONE-D-23-31531R2 

PLOS ONE

Dear Dr. Kern, 

I'm pleased to inform you that your manuscript has been deemed suitable for publication in PLOS ONE. Congratulations! Your manuscript is now being handed over to our production team.

Kind regards, 

on behalf of

Professor Mireia Jofre-Bonet 

Academic Editor

PLOS ONE